# Actin-Binding Proteins in Cardiac Hypertrophy

**DOI:** 10.3390/cells11223566

**Published:** 2022-11-11

**Authors:** Congbin Pan, Siqi Wang, Chao Liu, Zhanhong Ren

**Affiliations:** 1Hubei Key Laboratory of Diabetes and Angiopathy, Medicine Research Institute, Xianning Medical College, Hubei University of Science and Technology, Xianning 437100, China; 2School of Pharmacy, Xianning Medical College, Hubei University of Science and Technology, Xianning 437100, China

**Keywords:** actin-binding proteins, cardiac hypertrophy, F-actin, fetal genes

## Abstract

The heart reacts to a large number of pathological stimuli through cardiac hypertrophy, which finally can lead to heart failure. However, the molecular mechanisms of cardiac hypertrophy remain elusive. Actin participates in the formation of highly differentiated myofibrils under the regulation of actin-binding proteins (ABPs), which provides a structural basis for the contractile function and morphological change in cardiomyocytes. Previous studies have shown that the functional abnormality of ABPs can contribute to cardiac hypertrophy. Here, we review the function of various actin-binding proteins associated with the development of cardiac hypertrophy, which provides more references for the prevention and treatment of cardiomyopathy.

## 1. Introduction

Heart failure is one of the leading causes of morbidity and mortality globally [1]. Cardiac hypertrophy is the early pathological structural feature of heart failure and is usually observed in hypertension, aortic stenosis, diabetic cardiomyopathy and other cardiovascular diseases. Pathological cardiac hypertrophy is initially identified by a decrease in ventricular chamber size with an increase in wall thickness (concentric hypertrophy), in which the thickness of cardiomyocytes is usually increased. Pathological cardiac hypertrophy induces ventricular chamber dilatation (eccentric hypertrophy) accompanied by impaired systolic function (maladaptive remodeling) and lengthening of cardiomyocytes [2]. It is characterized by enhanced fibrosis, enlarged cardiomyocytes, cell death, mitochondrial dysfunction, oxidative stress, reactivation of fetal gene expression and dysregulation of Ca^2+^-handling proteins (Figure 1), which result from various mechanical stresses, genetic stimuli and neurohumoral mechanisms [2,3,4].

The research into the molecular mechanism of cardiac hypertrophy plays an important role in the prevention and treatment of cardiomyopathy, which has received growing attention in recent years. The E3 ligase tripartite motif-containing protein 16, for example, was found to effectively restrain the development of cardiac hypertrophy via mitigation of oxidative stress in cardiomyocytes [5]. Neuraminidase 1, also known as sialidases, can promote cardiac hypertrophy by reactivating the expression of fetal genes [6]. The histidine triad nucleotide-binding protein 1 attenuates cardiac hypertrophy via suppressing the TGF-β signaling pathway [7]. Despite numerous previous studies on cardiomyocyte morphogenesis, this notable process of cardiac hypertrophy remains poorly understood. 

It has been shown that cell morphogenesis is closely linked to the microfilament cytoskeleton [8]. The microfilament cytoskeleton is mainly composed of actin and actin-binding proteins (ABPs). Actin is one of the most abundant cytoskeletal proteins in eukaryotes and is involved in cell morphology change, migration, division and other cellular processes [9,10]. Actin takes two forms in cells: actin monomers (also known as globular actin, G-actin) and actin filaments (also known as filamentous actin, F-actin). Actin dynamics are finely regulated by a variety of ABPs (Table 1) [11]. Actin is involved in the formation of sarcomeres in cardiomyocytes [12]. The straight and uniform sarcomeric F-actin is critical for the contractile function of muscle [13]. In addition, actin assembly is thought to be related with autophagy [14,15]. The inhibition of F-actin disassembly can suppress autophagosome formation [16]. Several studies have found that F-actin is significantly accumulated abnormally in hypertrophic cardiomyocytes [17,18,19]. The dysregulation of F-actin accumulation may lead to cardiac hypertrophy through disrupting autophagy and sarcomeric structure. The function of ABPs in the development of cardiac hypertrophy has been gradually elucidated. Based on this, we briefly review the recent research progress on the various ABPs associated with cardiac hypertrophy, which has provided new strategies and targets for treating and reversing pathological hypertrophy. 

## 2. ABPs in Cardiac Hypertrophy

### 2.1. Profilin-1

Profilin is widely expressed in most eukaryotes and has a molecular weight of about 17 kDa [39]. There are various profilin isoforms expressed in different tissues. Profilin-1 is universally expressed, profilin-2 is specifically expressed in the brain and profilin-3 and profilin-4 are specifically expressed in kidney and testis, respectively [40]. Profilin accelerates the nucleotide exchange of G-actin and delivers ATP-G-actin to the growing barbed ends of F-actin through interacting with the poly-proline motifs of formin, vasodilator-stimulated phosphoprotein (VASP) and CDC42-activated Wiskott Aldrich syndrome protein (WASP)/WASP family [20,41,42,43]. 

Profilin-1 is directly associated with cardiac hypertrophy [44]. Overexpression of profilin-1 in the vascular tissues of FVB/N mice leads to vascular remodeling and hypertension by increasing actin aggregation, which provides mechanical stress for the development of cardiac hypertrophy [45,46]. It has been shown that the protein level of profilin-1 is significantly increased in mammalian hypertrophic hearts (Figure 2). The myocardin-related transcription factor megakaryoblastic leukemia (MKL) induces the expression of the signal transducer and activator of transcription 1 (STAT1) via its SAP-domain (SAF-A/B, acinus and PIAS protein domain) activity, which upregulates *PFN* expression [47]. Whether this is the explanation for the increased protein level of profilin in cardiac hypertrophy remains to be investigated. In cardiomyocytes, the functional abnormality of profilin-1 can change the abundance or activity of multiple proteins associated with cardiomyopathy. For example, the overexpression of profilin-1 can contribute to decreases in the phosphorylation level of endothelial nitric oxide synthases (eNOS) at Ser1177 in the hearts of spontaneous hypertensive rats [17]. Levels of atrial natriuretic peptide (ANP), brain natriuretic peptide (BNP), skeletal muscle α-actin (α-SMA) and phosphorylated ERK1/2 (active form) were significantly increased in neonatal rat ventricular myocytes (NRVMs) following stimulation by phenylephrine or endothelin 1, which can be inhibited by siRNA-directed *PFN1* silencing [44]. Increased phosphorylation of ERK1/2 activates the mechanistic (mammalian) target of rapamycin complex 1 (mTORC1) that subsequently inhibits autophagy [48,49,50]. It may be a potential key mechanism of cardiac hypertrophy mediated by the dysregulation of profilin-1 (Figure 2). Additionally, the inhibition of Rho-associated coiled-coil-containing protein kinase pathway (ROCK) can suppress the upregulation of profilin-1 induced by advanced glycation end products (AEGs) in H9c2 cells [51]. By comparison, overexpression of *PFN1* results in the reactivation of fetal genes (*NPPA* and *NPPB*), an increase in F-actin in myocardium and destruction of myofibrils [44]. These processes can be reversed by inhibiting the expression of profilin-1 [17]. The inhibition of profilin-1 expression in H9c2 cells and Sprague–Dawley rats can attenuate cardiac hypertrophy induced by AEGs [51,52]. In *Drosophila*, myocyte-specific overexpression of profilin leads to disorders in muscle fibers and sarcomeres, which result in damaged muscle ultrastructure and function [44]. 

### 2.2. ADF/Cofilin

Actin-depolymerizing factor (ADF)/cofilin consists of a single ADF homologous domain and has a molecular weight of about 15 kDa. The ADF/cofilin family contains ADF (also known as destrin, mainly expressed in endothelial and epithelial cells) and two cofilin isoforms (cofilin-1 is universal and cofilin-2 is cardio-specific) [53,54]. ADF/cofilin can bind to both G-actin and F-actin and can sever and depolymerize F-actin in regulating actin dynamics, which contributes to the cell contractility power [55]. The activity of cofilin is regulated by phosphorylation primarily from the ROCK/Lin-11, Isl1 and MEC-3 domain kinase (LIMK)/cofilin signaling pathway (Figure 3) [56,57]. Cofilin is inactivated via phosphorylation.

The abundance change in cofilin-2 does not play a role in the morphogenesis of neonatal rat cardiomyocytes [58], while its activity is closely associated with the development of cardiac hypertrophy. The levels of phosphorylated cofilin-2 are increased in myocardial hypertrophy through the activation of LIM-kinase (LIMK) by ROCK, which is induced by multiple neurohumoral factors, such as angiotensin II [59,60], endothelin 1 [19] and leptin [18,61]. In hypertrophic cardiomyocytes, the increase in levels of phosphorylated cofilin-2 results in an increase in F-actin/G-actin ratios and the levels of phosphorylated ERK1/2 and p38 [19,61,62,63,64]. Y-27632 [19], an inhibitor of ROCK, can reduce the levels of phosphorylated cofilin-2 through the inhibition of ROCK activity, which attenuates endothelin-1-induced neonatal cardiomyocyte hypertrophy, whereas this is achieved in ginseng (*Panax quinquefolius*) [62] through inhibition of p115Rho guanine nucleotide exchange factor (GEF) activity, which inhibits leptin-induced cardiac hypertrophy. In addition, WD-repeat domain 1 (WDR1), a major cofactor of the ADF/cofilin, has been reported to protect myocardium from myocardial hypertrophic stimuli [13].

### 2.3. Formin

Formin is a type of multidomain protein consisting of 7 subfamilies and 15 members in human genes. Formins are characterized by the presence of two conserved domains: formin homology 1 (FH1) and FH2. FH1 binds to the profilin–actin complex via poly-proline sequences and brings the G-actin to FH2, which promotes actin nucleation and polymerization [11,65].

#### 2.3.1. mDia1

mDia1 (mammalian homologue of *Drosophila* diaphanous 1) is an important member of the formin family. The intramolecular interaction between the N-terminal FH3 and C-terminal diaphanous autoregulatory domain (DAD) can induce mDia1 autoinhibition. Activation occurs when its N-terminal GTPase-binding domain (GBD) interacts with active Rho or Rac (Figure 4) [24,66]. 

Self-oligomerization of the mDia1 FH2 domain is essential for activation of serum response factor (SRF), which may function in the induction of cardiac responses to pressure overload [67,68]. Myocardial hypertrophy induced by the transverse aortic constriction (TAC) was attenuated in mDia1 knockout mice. The mDia1KO mice exhibited more severe dilation, fibrosis and higher mortality [68]. The molecular mechanism through which mDia1 regulates cardiac hypertrophy remains unclear and requires elucidation through further research.

#### 2.3.2. FHOD3 

FHOD3 (formin homology 2 domain-containing 3), a member of the formin family, is highly expressed in skeletal muscle and myocardium [69]. It is located in the thin actin filaments of the sarcomere and has been verified to be critical for sarcomere assembly, heart growth, development and functional maintenance [70,71]. The activation of FHOD3 requires the C-terminal phosphorylation induced by ROCK [72].

The expression and phosphorylation of FHOD3 is increased in cardiomyocytes purified from Angiotensin II-induced rat cardiac hypertrophy models, while the activation of FHOD3 inhibited by Y27632 attenuates Angiotensin II-induced cardiomyocyte hypertrophy [73]. Overexpression of the phosphomimetic mutant FHOD3-DDD results in cardiomyocyte hypertrophy in cultured neonate rat cardiomyocytes [73]. Additionally, *FHOD3* depletion in the neonatal mice heart induces disruption of the sarcomere structure and leads to lethality [74] while it does not cause any detectable changes in the sarcomere structure in the adult heart [74]. Furthermore, it was shown that the *Fhod3* variant increases the risk of the clinically apparent hypertrophic cardiomyopathy (HCM) [75].

### 2.4. CapZ

CapZ, a type of capping protein, anchors F-actin to the Z disc and regulates actin turnover, which contributes to sarcomere structural changes [76,77]. PIP2, a downstream effector of RAC1, can promote the dissociation of CapZ from F-actin by weakening their binding affinity [78,79,80].

Overexpression of CapZ in transgenic mice can lead to fatal cardiac hypertrophy [77]. It has been shown that hypertrophic agonists, phenylephrine or endothelin can reduce the binding affinity between CapZ and F-actin via PIP2-dependent pathways in NRVMs [81]. This may result in sarcomere remodeling, which induces cardiac hypertrophy. The cyclic mechanical strain activates downstream focal adhesion kinase (FAK) via the mechanotransduction of integrin, which then activates phosphatidylinositol 4-phosphate 5-kinase (PIP5K) through the RhoA/ROCK pathway. PIP5K phosphorylates phosphatidylinositol 4-phosphate (PI4P) in order to produce PIP2, which reduces the affinity of CapZ and F-actin binding, which contributes to the dysregulation of F-actin assembly and cardiac hypertrophy (Figure 5) [80,82,83,84].

### 2.5. Gelsolin

Gelsolin is responsible for F-actin severing, nucleating, bunding or capping [29]. The activation of gelsolin is regulated by PIP2 and Ca^2+^ [85,86]. In myocardial hypertrophy, the protein level of gelsolin is abnormally upregulated [87]. It has been shown that the phosphorylation of GATA4 is significantly suppressed by siRNA-directed *GSN* silencing, which inhibits cardiomyocyte hypertrophy [87]. In addition, decreased gelsolin can effectively inhibit the reactivation fetal gene (*NPPA* and *NPPB*) [88,89] and myocardial hypertrophy induced by palmitate or phenylephrine [90]. Cardiomyocyte hypertrophy induced by gelsolin overexpression can be blocked with p38 inhibitors [87]. Myocardial hypertrophy caused by left anterior descending coronary artery ligation is ameliorated in gelsolin knockout mice (GSN^−/−^) [88]. In addition, caspase cleavage of the gelsolin fragment plays an important role in cardiac hypertrophy [90].

### 2.6. Human Heart LIM Protein 

Human heart LIM protein (hhLIM), an important member of the LIM family, promotes *NPPA* expression by cooperating with cardiac transcription factor Nkx2.5 [91]. hhLIM is a type of F-actin bundling protein, and its overexpression contributes to cardiac hypertrophy [28,92]. Its interaction with the transcription factor may lead to cardiac hypertrophy, but the molecular mechanism of its mediating cardiac hypertrophy remains elusive. In addition, several LIM family proteins such as muscle LIM protein (MLP) and LIM domain-binding 3 have also been reported to play important roles in cardiac hypertrophy [93,94]. 

### 2.7. Myosin 

Myosin, a type of motor proteins, can move along actin filaments and produce contractile force [38]. The expression of MYH7 is upregulated in pathological myocardial hypertrophy [5,6], while it has no change in physiological hypertrophy [95]. Gene variants of cardiac myosin including MYH7 and MYH6 are closely linked with HCM and dilated cardiomyopathy (DCM) [96,97]. Furthermore, a meta-analysis of 7675 HCM patients showed that HCM patients with MYH7 mutations had an earlier age of onset, resulting in a more severe phenotype [98]. *MYH7* p.Val320Met contributes to the increased risk of sudden cardiac death of hypertrophic cardiomyopathy [99]. It is worth noting that small molecule drugs, such as mavacamten and aficamten, targeting myosin have been employed in clinical trials for treatment of HCM [97,100,101]. However, the molecular mechanism by which the gene mutation of myosin mediates the cardiac hypertrophy remains unknown.

### 2.8. Dystrophin

Dystrophin can bind to F-actin through its N-terminal actin-binding domain 1 (ABD1) and ABD2, which is strengthened by its C-terminal region [22,102]. Duchenne muscular dystrophy (DMD), closely linked with the altered expression or null mutation of dystrophin in cardiac and skeletal muscles, are frequently complicated by cardiac hypertrophy and dilated cardiomyopathy [103,104]. In addition, the transition of compensated cardiac hypertrophy to heart failure is accompanied by expression decrease in the dystrophin [105]. The rescued dystrophin can prevent and attenuate cardiac hypertrophy in a DMD mouse model induced by a truncation mutation of dystrophin [106]. Preserving dystrophin can attenuate hypertensive eccentric cardiac hypertrophy [107]. This suggests that dystrophin may be an effective therapeutic target.

### 2.9. Other ABPs

Other ABPs have been reported to associate with cardiovascular diseases. For example, the protein level of FHOD1 is significantly increased in DCM [108]. Knockdown of leiomodin 2, a capping protein, results in shorter filaments, which subsequently leads to DCM and mortality in infancy [109]. Cyclase-associated protein 2, a protein that regulates thin filament length by sequestering G-actin and severing F-actin [12], is associated with cardiomyopathy [110]. Tropomodulin plays an important role in DCM [111]. In addition, gene mutations of structural components of sarcomeres such as titin, troponin C, troponin I and troponin T, most commonly involve DCM [97,112]. Determination of the function and molecular mechanism of ABPs in the development of cardiac hypertrophy requires further research.

## 3. Concluding Remarks

Cardiac hypertrophy is a common prepathology of heart failure, which is caused by multiple pathological stimuli [7]. It is characterized by enlarged cardiac myocytes, increased fibrosis, reactivation of fetal genes and sarcomere remodeling [4] as a result of actin dynamics regulated by ABPs. Therefore, ABPs in cardiomyocytes may be a prospective therapeutic target for cardiac hypertrophy and heart failure. We briefly summarize the reported ABPs associated with myocardial hypertrophy in Table 2 for reference in the design of drugs for cardiomyopathy treatment. In addition, it is worth studying the molecular mechanism through which other ABPs (Table 1) mediate cardiac hypertrophy, which provides more choices of the therapeutic targets for cardiomyopathy.

## Figures and Tables

**Figure 1 cells-11-03566-f001:**
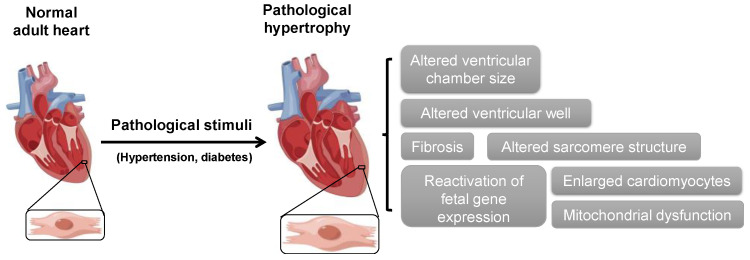
**General features of pathological hypertrophy.** Examples of pathological stimuli (such as hypertension and diabetes) responsible for cardiac hypertrophy.

**Figure 2 cells-11-03566-f002:**
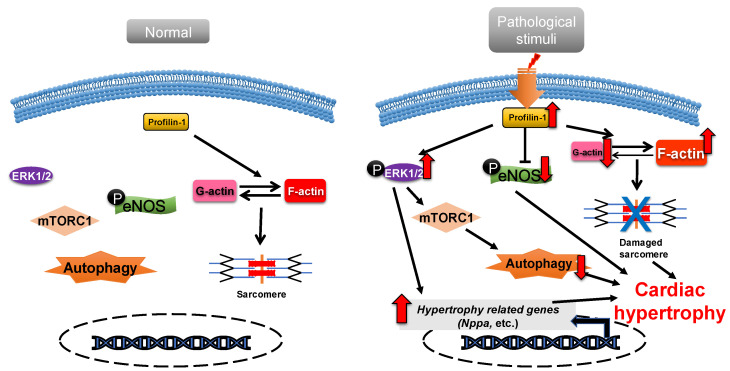
**Profilin-1 mediates cardiac hypertrophy.** In normal cardiomyocytes, profilin-1 is at a basal level and the fetal genes are not activated. Pathological stimuli increase the protein level of profilin-1, which results in ERK1/2 activation, F-actin accumulation and eNOS inhibition. This results in the reactivation of hypertrophy-related genes, inhibition of autophagy and damage to sarcomere structure and, ultimately, the development of cardiac hypertrophy.

**Figure 3 cells-11-03566-f003:**
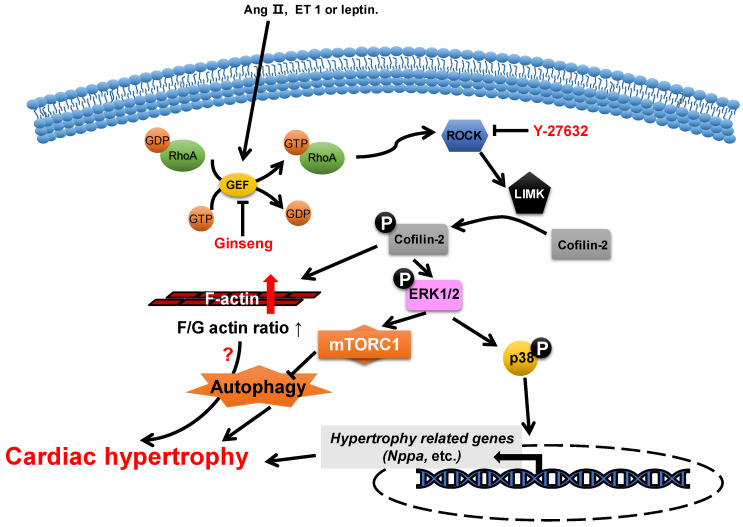
**Proposed roles of cofilin-2 in cardiac hypertrophy.** Neurohumoral factors (e.g., Ang II, ET 1 and leptin) lead to cofilin-2 phosphorylation through the RhoA/ROCK/LIMK signaling pathway. Phosphorylated cofilin-2 can lead to F-actin accumulation, which may subsequently contribute to cardiac hypertrophy through disrupting autophagy. In addition, it promotes the activation of ERK1/2 and p38, which contributes to the inhibition of autophagy and the reactivation of hypertrophy-related genes, which subsequently cause cardiac hypertrophy.

**Figure 4 cells-11-03566-f004:**
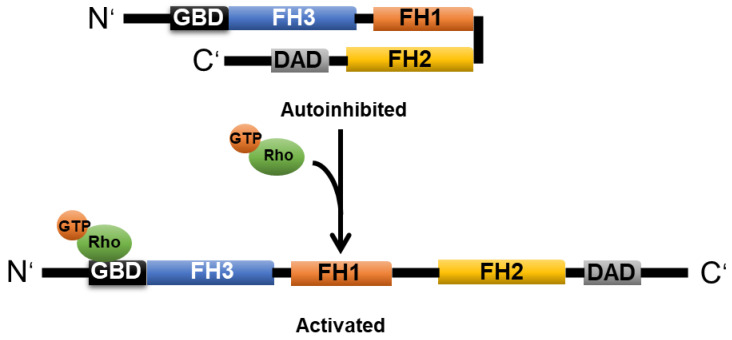
**Domain structure and activation of mDia1.** The interaction between FH3 and DAD causes mDia1 to remain in a closed and inactive conformation. GTP-loaded Rho can activate mDia1 through binding to the N-terminal GBD, which causes mDia1 to adopt an open and active form. The illustrated molecules are not drawn to scale.

**Figure 5 cells-11-03566-f005:**
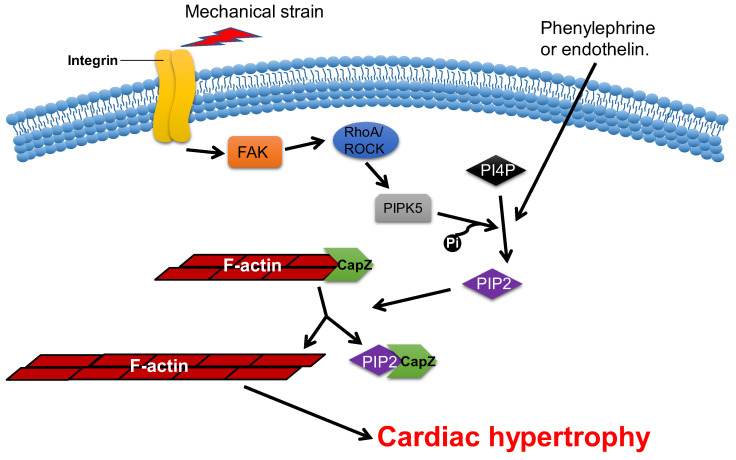
**CapZ regulates cardiac hypertrophy.** Mechanotransduction leads to the activation of RhoA/Rho-kinase pathway through integrins, which reduce the binding affinity of CapZ and F-actin. It subsequently causes cardiac hypertrophy.

**Table 1 cells-11-03566-t001:** Actin-binding proteins.

Types	ABPs	Basic Function	Refs.
G-actin-binding	Profilin, thymosin β4, cofilin	Bound to G-actin	[11,20,21]
F-actin-binding	Dystrophin, tropomyosin	Bound to F-actin	[9,11,22]
Actin-nucleating	Formin, Arp2/3 complex, proteins with tandem WH2 domains, leiomodin	Nucleation to initiate actin polymerization	[11,23,24,25]
Actin-elongating	Formin, tetramers of Ena/VASP	Regulation of actin assembly	[11,24]
Actin-bundling	Fimbrin/Plastin, hhLIM, gelsolin	Causes parallel F-actin filaments to closely pack together	[26,27,28,29]
Severing	ADF/cofilin, gelsolin, twinfilin, FRL-α, INF-2	Severs F-actin	[30,31,32,33,34]
Capping	Twinfilin, gelsolin, tropomodulin, CapZ, Arp2/3 complex	Caps F-actin to inhibit actin polymerization	[11,35,36,37]
Motor	Myosin	Cargo transfer	[38]

**Table 2 cells-11-03566-t002:** Inductive cues for cardiac hypertrophy.

ABPs	Function	Protein Synthesis/Phosphorylation		Refs.
		Over Expression	Knock Down/Out	
Profilin-1	Polymerization	ANP, BNP and α-SMA↑; p-eNOS↓.	ANP, BNP and p-ERK1/2↓; p-eNOS↑.	[17,44,51,52]
Cofilin-2	Severing	/	/	[11]
mDia1	Nucleation	/	SRF, MRTF, pERK1/2 and pFAK↓	[68,113]
FHOD3	Nucleation	/	ANP, BNP and MYH7↑.	[69,74]
CapZ	Capping	/	/	[77]
Gelsolin	Severing, Capping	/	ANP and BNP↓.	[85,89,114]
hhLIM	Bundling	/	BNP and α-SMA↓.	[28,92]

↑ Increased, ↓ decreased, / no reports.

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
