# Peer review of "Actin-Binding Proteins in Cardiac Hypertrophy"

_cells, 2022, doi:10.3390/cells11223566_

Round 1

Reviewer 1 Report

The content of this manuscript is potentially interesting. Unfortunately,  the manuscript is hard to read (totally unreadable in some sections) because of language inadequacy. It is not a matter of subtelties or typos, structure and syntaxis are simply not there; faults are practically in every sentence, too numerous  to be detailed.

Before the manuscript can be reviewed for its scientific content, it should be entirely rewritten in English. 

Author Response

Response to Reviewer 1 Comments

Point 1: The content of this manuscript is potentially interesting. Unfortunately, the manuscript is hard to read (totally unreadable in some sections) because of language inadequacy. It is not a matter of subtelties or typos, structure and syntaxis are simply not there; faults are practically in every sentence, too numerous to be detailed.

Response 1: We thank the reviewer for this suggestion. The revised manuscript has undergone English language editing by MDPI (https://www.mdpi.com/authors/english). The related language editing is marked in blue in the revised manuscript.

Reviewer 2 Report

The Review article by Pan et al on Actin binding protein in Cardiac hypertrophy discusses the role of actin binding proteins role in the cardiac hypertrophy.  Authors first reviewed about the hypertrophy pathology and ABP’s association in cardiac hypertrophy. Then authors explained each ABP function and its dysregulation in cardiac hypertrophy. However, authors need to address the following concerns before publication.

Major comments

1.       Human heart LIM protein (hhLIM) is novel F-actin binding protein functions actin bundling protein. What is the role of actin bundling proteins in cardiac hypertrophy?

2.       The cross-talk of cardiac transcription factors with actin binding proteins and its role in cardiac hypertrophy should be included in the manuscript.

3.       To date 12 different actin gene mutations have been discovered in patients with hypertrophic cardiomyopathy. How does mutations in ABPs leading to hypertrophy?

4.       Rho signaling pathway plays major in cardiac hypertrophy through various RHO dependent downstream mechanisms. How does Rho/Rho kinase signaling stabilizes the ABPS and dysregulation impact in cardiac pathology?

Minor comments

Authors should update latest references in the manuscript.

Author Response

Response to Reviewer 2 Comments

The Review article by Pan et al on Actin binding protein in Cardiac hypertrophy discusses the role of actin binding proteins role in the cardiac hypertrophy. Authors first reviewed about the hypertrophy pathology and ABP’s association in cardiac hypertrophy. Then authors explained each ABP function and its dysregulation in cardiac hypertrophy. However, authors need to address the following concerns before publication.

Major comments

Point 1: Human heart LIM protein (hhLIM) is novel F-actin binding protein functions actin bundling protein. What is the role of actin bundling proteins in cardiac hypertrophy?

Response 1: We thank the reviewer for this critical point. We have added new contents about roles of actin bundling proteins such as gelsolin and hhLIM in cardiac hypertrophy in the revised manuscript. (Page 9, line 195, 197-198, 200-201; page 9, 205-212)

Point 2: The cross-talk of cardiac transcription factors with actin binding proteins and its role in cardiac hypertrophy should be included in the manuscript.

Response 2: We appreciate this reviewer so much for this wonderful suggestion. We have added new contents and disscussions about the association of cardiac transcription factors with actin binding proteins and its role in cardiac hypertrophy in the revised manuscript. (Page 3, line 74-78; page 7, line 144-145; page 9, line 197-198, 206-207)

Point 3: To date 12 different actin gene mutations have been discovered in patients with hypertrophic cardiomyopathy. How does mutations in ABPs leading to hypertrophy?

Response 3: The reviewer pointed out the interesting question. We have added new contents about mutations in the ABPs associated with hypertrophic cardiomyopathy and dilated cardiomyopathy in the revised manuscript. The molecular mechanism by which mutations in ABPs mediating hypertrophy requires further studies. (Page 7-8, line 163-171; page 9-10, line 216-223, 226-228, 230-231, 241-242)

Point 4: Rho signaling pathway plays major in cardiac hypertrophy through various RHO dependent downstream mechanisms. How does Rho/Rho kinase signaling stabilizes the ABPS and dysregulation impact in cardiac pathology?

Response 4: We thank this reviewer for this critical point and have discussed the roles of Rho/Rho kinase signaling pathway mediating ABPs in cardiac hypertrophy in the revised manuscript. (Page 3, line 68-69; page 4, line 87-89; page 5, line 109-112, 115-117, 120-125; page 7, line 159-160, 163-166;  Page 8, line 176, 182-187)

Point 5: Minor comments

Authors should update latest references in the manuscript.

Response 5: We are grateful to this reviewer for this suggestion. We have update latest references in the revised manuscript. (Marked in blue)

Reviewer 3 Report

In their review the authors describe the role of a subgroup of actin binding proteins in cardiac hypertrophy. They highlight the importance of the topic and appropriate figures and tables to facilitate an overview. It remains unclear why they chose the 7 ABPs mentioned in this review. A clear mechanistic link is often missing but rather associations are pointed out. A few misleading comments are made and important details are missing or inprecise. Overall, this is an interesting topic but the review needs major revision:

It starts with a misleading description of what hypertrophy is:

Intro: definition of hypertrophy would help, and separating it from its association to fibrosis, etc. The subtitle of figure 1 suggests a misunderstanding: an enlarged cardiomyocyte is not a feature, especially not a “most significant” feature but part of its definition. Maybe also provide a clinical definition.

Figure 1: “impaired protein” is a vague description – what does this mean? Impaired function? Degradation? Aggregation? This could be explained in detail in the text 

Line 35: morphogenesis is determined by microfilament cytoskeleton; separate from this idea: hydraulic pressure and mechanical force are potential influencer; putting these 3 together in a list is not logical. They are on a different mechanistic level

Line 56: eNOS S1177 can be a result of profilin-1 overexpression, but there are other causes as well. The wording is not clear.

Figure 2: shorten title

Line 78: reference missing

Line 128: Figure 5 description should be provided in a short and precise manner

Line 69+90+124: maybe the authors can elaborate why F-actin accumulation leads to cardiac hypertrophy: is this due to aggregation of dysfunctional proteins? Failing autophagy?

Concluding remarks:

The authors conclude that the few ABPs mentioned in this review may be a therapeutic target for cardiac hypertrophy. The authors miss to describe the plethora of ABPs that are not described in detail here like integrins, MLC, titin, troponin, CamKII, Dystrophin, Hsp, and many more that exist and already have shown to have major therapeutic implications.

Kind suggestion: maybe it would also be interesting to add the role of actin and ABPs in physiological hypertrophy and its similarities/differences to pathological hypertrophy if data is available

English: major revision needed

Line 15: associated

Line 22: characterized by

Line 28: English can be improved …

Line 35: related to 

Line 37: involves in – is involved in 

Line 38: contain – might be the wrong verb

Line 54: shown

Line 56: cardiomyopathy

Line 63: glycation+

Line 72: consists of 

Line 74: is (not was)

Line 79: content change …. change of protein abundance? 

Line 84: attenuates ?

Line 83-86: sentence is not understandable

Line 93: “formin is a kind of …” very colloquial English

Line 95: promotes

Line 101-102: attenuated in mDia1 knockout mice.

Line 131: it has been shown

Line 132: fetal gene reactivation

Line 144: caused by multiple pathological stimuli…

Line 145: -which

Author Response

Response to Reviewer 3 Comments

In their review the authors describe the role of a subgroup of actin binding proteins in cardiac hypertrophy. They highlight the importance of the topic and appropriate figures and tables to facilitate an overview.

Point 1: It remains unclear why they chose the 7 ABPs mentioned in this review.

Response 1: We thank this reviewer for this critical point. ABPs mainly discussed in this revised manuscript are listed separately based on their functions in regulating actin dynamics. We have chosen 1-2 representative ABPs from the same functional type to elaborate in this revised manuscript.

Point 2: A clear mechanistic link is often missing but rather associations are pointed out.

Response 2: We are grateful to this reviewer for this point. Different ABPs may have distinct roles in myocardial hypertrophy. For example, the overexpression of profilin 1 or CapZ can lead to cardiac hypertrophy while gene mutations in MYH7 and MYH6 is associated with hypertrophic cardiomyopathy and dilated cardiomyopathy. However, whether gene mutations in MYH7 and MYH6 induce cardiac hypertrophy or not requires further studies. Therefore, the elaboration on ABPs is based on their research progress in the revised manuscript.

Point 3: A few misleading comments are made and important details are missing or inprecise.

 Response 3: We have corrected the relevent contents in the revised manuscript as suggested.

Overall, this is an interesting topic but the review needs major revision:

Point 4: It starts with a misleading description of what hypertrophy is:

Intro: definition of hypertrophy would help, and separating it from its association to fibrosis, etc. The subtitle of figure 1 suggests a misunderstanding: an enlarged cardiomyocyte is not a feature, especially not a “most significant” feature but part of its definition. Maybe also provide a clinical definition.

Response 4: We thank the reviewer for this good suggestion. We have removed "The enlarged cardiomyocyte is one of the most significant structural features of cardiac hypertrophy" from the legend of revised Figure 1 (Page 2, line 33) and added the clinical definition in the revised manuscript (Page 1, line 23-27) and revised Figure 1.

 Point 5: Figure 1: “impaired protein” is a vague description – what does this mean? Impaired function? Degradation? Aggregation? This could be explained in detail in the text.

Response 5: We thank the reviewer for pointing this mistake of description. We have removed the vague description of "impaired protein" in the revised Figure 1.

Point 6: Line 35: morphogenesis is determined by microfilament cytoskeleton; separate from this idea: hydraulic pressure and mechanical force are potential influencer; putting these 3 together in a list is not logical. They are on a different mechanistic level

Response 6: We agree with the reviewers and have removed the description of hydraulic pressure and mechanical force from the relevent section in the revised manuscript. (Page 2, line 44)

Point 7: Line 56: eNOS S1177 can be a result of profilin-1 overexpression, but there are other causes as well. The wording is not clear.

Response 7: We have rephrased this statement in the revised manuscript. (Page 4, line 80-83)

Point 8: Figure 2: shorten title

Response 8: We have shortened the title of the revised Figure 2.

Point 9: Line 78: reference missing

Response 9: We have added the related reference in the revised manuscript. (Page 5, line 109)

Point 10: Line 128: Figure 5 description should be provided in a short and precise manner

Response 10: We are grateful to the reviewer for this critical point. We have rewritten the relevant description in the revised manuscript. (Page 9, line 189-192)

Point 11: Line 69+90+124: maybe the authors can elaborate why F-actin accumulation leads to cardiac hypertrophy: is this due to aggregation of dysfunctional proteins? Failing autophagy?

Response 11: We appreciate this reviewer so much for this wonderful suggestion. Relevant content has been added in the revised manuscript. (Page 2, line 50-55)

Point 12: Concluding remarks:

The authors conclude that the few ABPs mentioned in this review may be a therapeutic target for cardiac hypertrophy. The authors miss to describe the plethora of ABPs that are not described in detail here like integrins, MLC, titin, troponin, CamKII, Dystrophin, Hsp, and many more that exist and already have shown to have major therapeutic implications.

Response 12: We thank this reviewer for this critical point and have added relevent contents in the revised manuscript. (Page 9-10, line 205-233, page 10, line 241-243)

Point 13: Kind suggestion: maybe it would also be interesting to add the role of actin and ABPs in physiological hypertrophy and its similarities/differences to pathological hypertrophy if data is available.

Response 13: We thank this reviewer for the good suggestion and have added the relevant section in the revised manuscript. (Page 9, line 215-216)

Point 14: English: major revision needed

Line 15: associated

Line 22: characterized by

Line 28: English can be improved …

Line 35: related to

Line 37: involves in – is involved in

Line 38: contain – might be the wrong verb

Line 54: shown

Line 56: cardiomyopathy

Line 63: glycation+

Line 72: consists of

Line 74: is (not was)

Line 79: content change …. change of protein abundance?

Line 84: attenuates ?

Line 83-86: sentence is not understandable

Line 93: “formin is a kind of …” very colloquial English

Line 95: promotes

Line 101-102: attenuated in mDia1 knockout mice.

Line 131: it has been shown

Line 132: fetal gene reactivation

Line 144: caused by multiple pathological stimuli…

Line 145: -which

Response 14: We are grateful to this reviewer for pointing these errors of English language. We have corrected them in the revised manuscript. (Page 1, line 15, line 27; page 2, line 34-36, line 43, line 46-47, line 47-48; page 3, line 73; page 4, line 80, line 93; page 5, line 104, line 107, line 113, line 122, line 120-125; page 6, line 134, line 137; page 7, line 147; page 9, line 197, line 199; page 10, line 246-247, line 249)

Round 2

Reviewer 2 Report

Accept

Author Response

Response to Reviewer 2 Comments

Point 1: Accept.

Response 1: We thank the reviewer for this decision.

Reviewer 3 Report

Much better. Regarding previous point 5: impaired autophagy should be added in the figure as a potential key mechanism of cardiac hypoertrophy and in the text (e.g. PMID 32393148 or others). 

English still needs editing: e.g.

line 43 linked to

line 197 -the

and others

Author Response

Response to Reviewer 3 Comments

Point 1: Much better. Regarding previous point 5: impaired autophagy should be added in the figure as a potential key mechanism of cardiac hypoertrophy and in the text (e.g. PMID 32393148 or others).

Response 1: We thank the reviewer for this suggestion. Relevant content has been added in the revised Figures (Figure 2 and 3) and the revised manuscript. (Page 2, line 52-53; page 3, line 56; page 4, line 89-91; page 5, line 106; page 6, line 135-138)

Point 2: English still needs editing: e.g.

line 43 linked to

line 197 -the

and others

Response 2: We are grateful to this reviewer for this point. We have corrected them in the revised manuscript. (Page 2, line 43; page 8, line176-177; page 9, line 198-199)
